# The Impact of Pollution on Cultural Heritage in the Historic Centre of Porto, Portugal

Fátima Matos Silva [1,2,*], Marta Arreiol [3] and Ana Fragata [4,5]

1   REMIT—Research on Economics, Management and Information Technologies, Department of Tourism, Heritage and Culture, Universidade Portucalense, 4200-072 Porto, Portugal
2   CITCEM—Transdisciplinary Research Centre Culture, Space and Memory, Faculty of Arts and Humanities, University of Porto, Via Panorâmica s/n, 4150-564 Porto, Portugal
3   Conservation and Restoration Clinic, Universidade Portucalense, 4200-072 Porto, Portugal; m4rtaa@gmail.com
4   GeoBioTec, Geosciences Department, University of Aveiro, Campus de Santiago, 3810-193 Aveiro, Portugal; afragata@ua.pt
5   Department of Tourism, Heritage and Culture, Universidade Portucalense, 4200-072 Porto, Portugal
*   Correspondence: mfms@upt.pt

**Abstract:** Pollution is a constant threat to cultural heritage, mainly affecting its constituent materials, and it is urgent to implement mitigation and adaptation measures to prevent pollution. The city of Porto currently has several initiatives that aim to prepare this municipality for climate change adaptation. This article aims to study the impact of pollution on built heritage, as well as the initiatives that are being implemented in the Municipality of Porto (as part of the Portuguese Camino to Santiago) to protect heritage, based on three case studies, namely Carmelitas Church, São João Novo Church, and Vitória Church, contextualising them over time and understanding their structure and materials. The methodology is based on an anomalies survey through local and surrounding photographic records to assess the effects of pollution, following the model developed at Carmo Church in Olinda, Pernambuco. This study's results showed that the stone facade of Carmelitas Church, which is in a busier area of the city, is much more deteriorated when compared with the other case studies due to the direct impact of pollution.

**Keywords:** impact of pollution; degradation of materials; places of worship; stone materials; tiles; Porto City





## 1. Introduction

Built along the Douro River, the city of Porto shows an urban landscape with an ancient history, where its historic centre is a singular cultural heritage testimony [1].

Cities are referred to as recent ecosystems on the face of the Earth, where human beings, animals, and other organisms cohabit the same space [2]. The imbalance in those cities occurs when certain factors of anthropomorphic or natural origin compromise and modify the surrounding environment, promoting environmental changes and significant impacts on urban ecosystems [3].

This investigation aims to study how these factors directly affect the built heritage associated with the Central Portuguese Way to Santiago based on three case studies: Carmelitas Church, São João Novo Church, and Vitória Church.

This article is divided into two parts: (i) the first one presents the historical and urban context of the city of Porto, as well as the environmental and climatic characterisation; (ii) the second part is related to the characterisation and analyses of the environment surrounding each of the case studies, identifying their degradation patterns of the exterior stone facades and relating them to the surrounding environment.

This work contributes to a better knowledge of the damage mechanisms of this cultural heritage in an urban context as an input to their preservation.

## 2. The Historic Centre of Porto

Cities, from the small urban clusters of antiquity to the large metropolises of today, have always performed economic, social, and cultural functions as spaces of continuous evolution. Their activities were mainly centred in their historic centres, where the oldest parts of cities are located with their cultural heritages of several eras [4]. Currently, despite having lost part of their attractive and polarising power, which did not happen with the historic centre of Porto, on the contrary, the symbolic image of those historic centres will always remain, and, as a result, there is a set of norms regarding their preservation and valorisation [4].

Built along the Douro River, the city of Porto has an urban landscape with an ancient history. Its historic centre contains a historical testimony of the passage of various eras, visible through its urban fabric of medieval origin or even through the remaining sections of the "Fernandina" defensive wall (Figure 1) that once surrounded the entire city [5].

As in other urban centres, the industrial and post-industrial eras brought great changes in the social, urban, and economic domains, causing the migration of the population and moving industrial production to the outskirts of cities [6].

During the 20th century, several plans emerged in Porto that were meant to address issues related to city planning [7]. In 1916, Cunha Morais developed a plan named "The Improvements of the City of Porto", which was based on urban strategies used in other European cities, such as Barcelona or Paris, where a reticulated mesh was overlaid on the pre-existing urban structure, resulting in a loss of importance of the historic centre. Later, in 1932, the plan "Prologue to the Porto City Plan" by Ezequiel de Campos aimed to harmonise areas of traditional occupation and also areas under development [7]. The strategies of the "Improvement Plan" in 1956 and the "Auzelle Plan" in 1962 allowed the historic centre of Porto city to be emptied, now with new urban planning [5].

At the same time, there was a clear state of degradation and desertification in this area, which led to the awareness of the importance of creating measures and mechanisms that would allow the historic centres to be added to the urban fabric so that they could be the object of joint intervention, as well as the development of actions to preserve and safeguard the historic heritage [6].

Important organisations, such as CRUARB (Commissariat for Urban Renewal of the Ribeira-Barredo Area), emerged to requalify and intervene in those spaces. The activities carried out by this organisation led to the recognition of the Historic Centre of Porto as a Cultural Heritage of Humanity in 1996 [7]. In 2003, Porto Vivo–SRU replaced CRUARB, although maintaining the downtown of Porto as the target of its intervention [7].

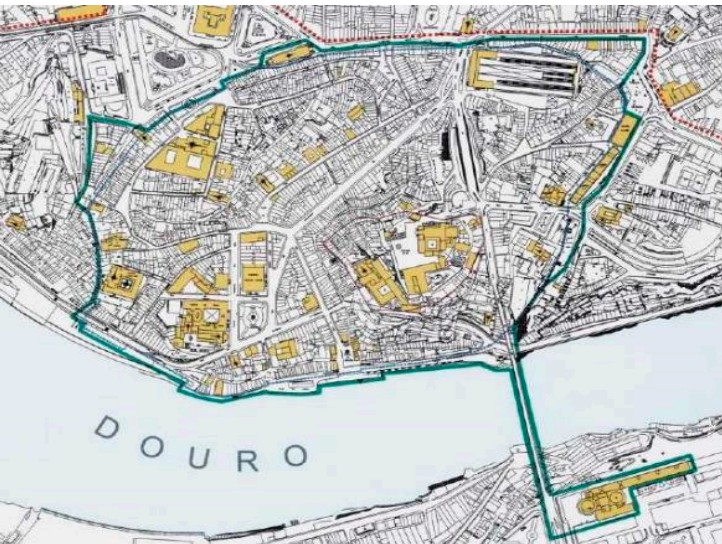

**Figure 1.** Historic Centre of Porto. The green line refers to delineating the area classified as a Cultural Heritage of Humanity [8].

### 3. Climatic and Environmental Characterisation in the City of Porto

*3.1. Analysis of Environmental Factors of Degradation in Porto's Built Heritage*

Most cities are on the front lines of climate change impacts. The imbalance in these urban ecosystems can occur from human origins, such as car traffic, industrial activity, compromising and modifying the surrounding environment, and promoting environmental changes and significant impacts. These changes may also result from naturally occurring events, which are subject to frequent natural cycles without any human intervention, such as emissions from volcanic eruptions or forest fires of natural origin, dust carried by the wind, emissions of organic compounds from plants, or sea spray [3].

Most chemical substances are often associated with the idea of pollutants, such as carbon or nitrogen, and exist in nature in non-toxic or polluting forms. Their toxicity only appears with increasing concentrations and/or combinations. Therefore, these substances may have a higher or lower impact on air quality (Figure 2) depending on their chemical compositions, concentrations in the air mass, and meteorological conditions [9].

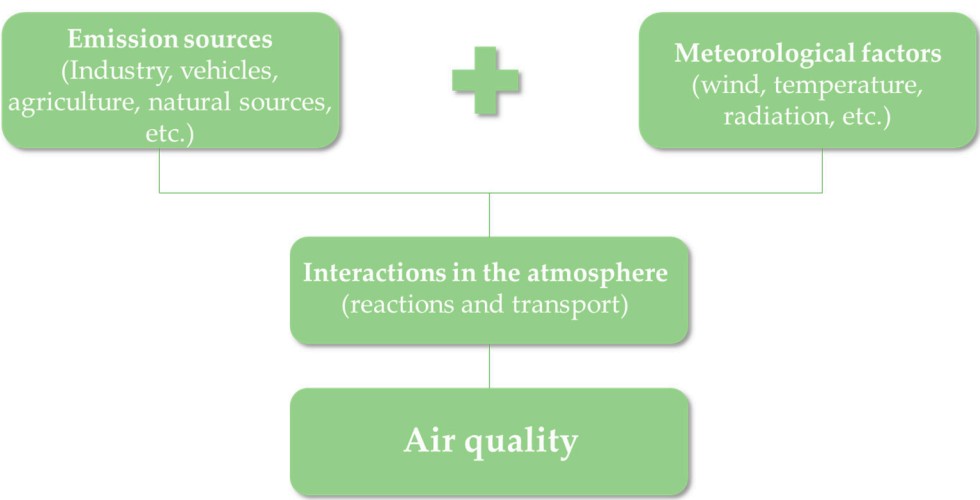

**Figure 2.** Factors that influence air quality. Source: APA.

In 2022, the Portuguese Environment Agency (APA) issued the State of the Environment Report—Portugal, an exercise carried out annually that monitors a set of factors that aim to provide a perspective of the state of the environment in Portugal, taking into consideration the goals and commitments associated with the environment and sustainable development [10].

After a careful analysis of the data collected in this report, it was observed that, between the years 2020 and 2023 (until August), the air quality index (IQAr) in the Municipality of Porto had maintained a "good" rating in this period. It was also observed that, from 2021 onwards, there was a small reduction in days classified as "very good" and a slight increase in days classified as "medium" and "weak", as we can see in Figure 3.

As there are no more recent data, the remaining analysis of the climate and environmental characterisation of the Municipality of Porto was based on the report prepared by the Portuguese Environment Agency entitled "Emissions of air pollutants by Municipality—2015, 2017 and 2019". This document, presented in 2021, is based on the international and community commitments assumed by Portugal, namely the Convention on Long-Range Transboundary Air Pollution, the Stockholm Convention on Persistent Organic Pollutants, and the United Nations Framework Convention on Climate Change [11].

In this article, the following pollutants are considered:

- Greenhouse gases (carbon dioxide ($CO_2$), methane ($CH_4$), nitrous oxide ($N_2O$), hydrofluorocarbons (HFC), perfluorocarbons (PFC), sulphur hexafluoride (SF6), and nitrogen trifluoride (NF3));

- Emissions of ozone precursors (volatile organic and non-methane compounds (NMVOCs)) and nitrogen oxide (NOx);
- Emissions of fine particles ($PM_{2.5}$ and $PM_{10}$);
- Emissions of acidifying gases (sulphur dioxide ($SO_2$), nitrogen oxides (Nox), and ammonia ($NH_3$)).

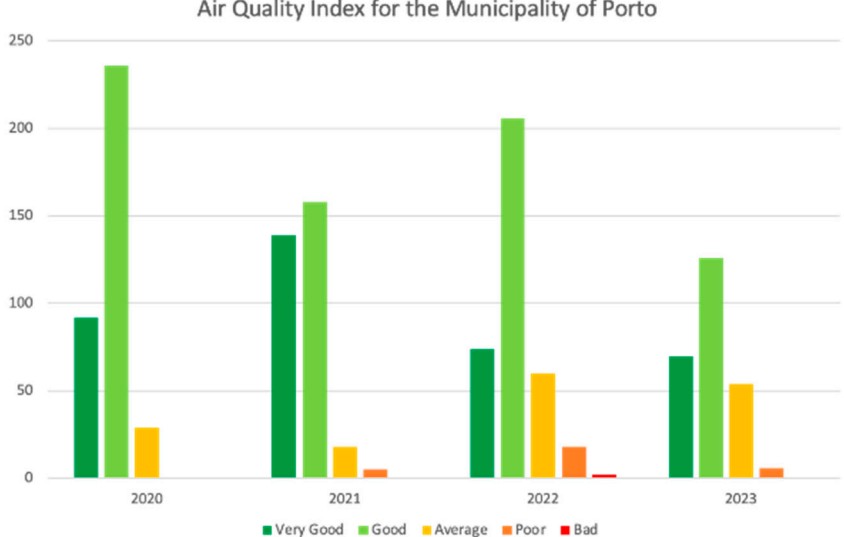

**Figure 3.** Air quality index in the Municipality of Porto between January 2020 and August 2023. Source: QualAR.

The results will be presented according to the categories included in the Guidelines for Reporting Emission Data Group. The NRF group represents 14 broad categories of emission sources established by the UNECE (United Nations Economic Commission for Europe) [12].

According to the Portuguese Environmental Agency, the total greenhouse gas emissions represented a total of 63.6 Mt $CO_2$, which is an increase of 8.1% between 1990 and 2019 [11]. In the Municipality of Porto, the APA indicated a slight decrease between 2015 and 2019 of 3.1%, with the Road Transport (F_RoadTransport) sector contributing the largest amount of emissions (Figure 4). The Industry (B_Industry) and Stationary Combustion (C_OtherStacionaryComb) sectors significantly reduced their emissions during this period.

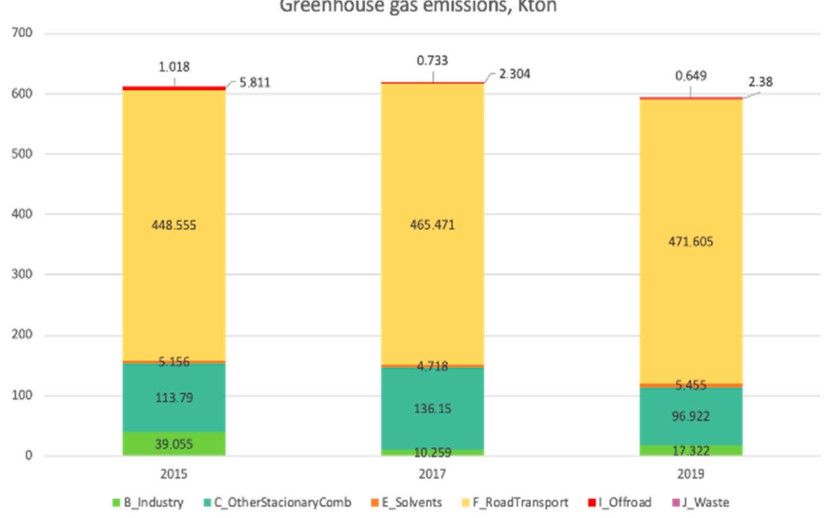

**Figure 4.** Greenhouse gas emissions in the Municipality of Porto for 2015, 2017, and 2019 by sector. Source: APA.

Regarding tropospheric ozone precursor substances (Figure 5), there was a 40% decrease between 1990 and 2019 in Portugal, corresponding to 317 Kt in 2019 [12]. In the Municipality of Porto, these emissions were characterised by a 10.8% reduction between 2015 and 2019, where the Road Transport sector (F_RoadTransport) registered the highest percentage of these emissions, and the industrial sector (B_Industry) exhibited the largest reduction (−0.229 Kt) (Figure 5).

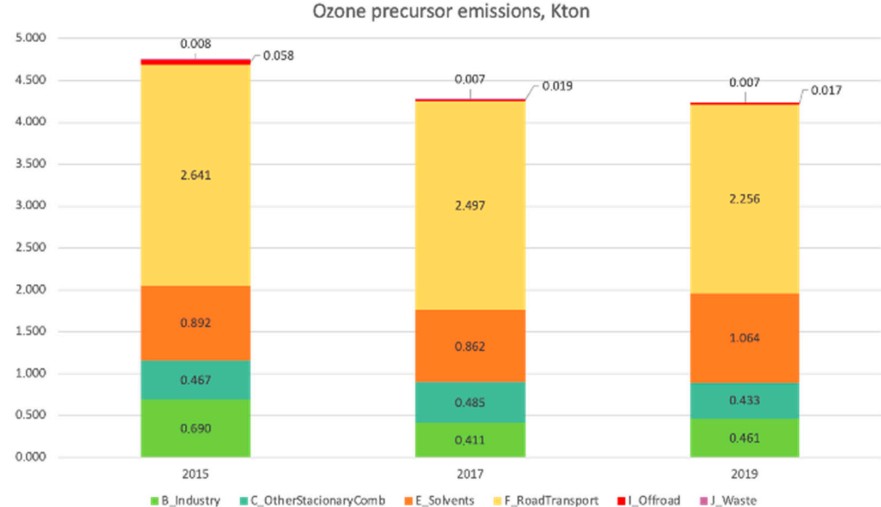

**Figure 5.** Emissions of ozone precursors in the Municipality of Porto for the years 2015, 2017, and 2019, by sector. Source: APA.

Between 1990 and 2019, acidifying gas emissions in Portugal fell by 64%, mainly in the industry, energy, and waste sectors, with reductions of 54%, 96% and 81% respectively [12]. For the Municipality of Porto, these substances decreased by around 16.8% between 2015 and 2019, again in the Road Transport sector (F_RoadTransport), contributing most of the emissions during this period (Figure 6). The greatest reduction in emissions was seen in the Industry sector (B_Industry), in the Stationary Combustion sector (C_Other StacionaryComb), and the Rail Transport sector (I_Offroad).

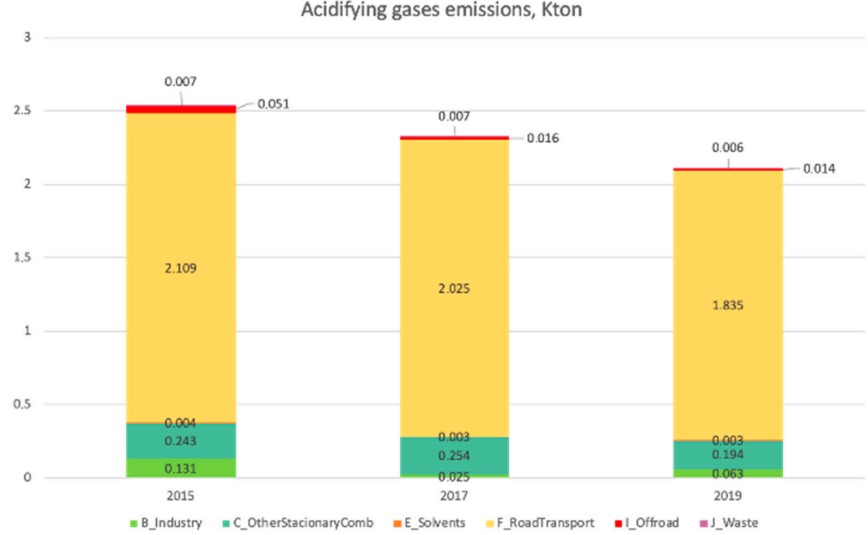

**Figure 6.** Emissions of acidifying gases in the Municipality of Porto for 2015, 2017 and 2019 by sector. Source: APA.

According to the Portuguese Environment Agency, the average concentration of suspended particles has exhibited a decreasing trend since the year 2000, with the val-

ues recorded in 2019 not exceeding the maximum annual limit of 35 days per year, according to Decree-Law No. 102/2010. For the Municipality of Porto, the 2020/2021 APA report indicated a 28.6% reduction in fine particles between 2015 and 2019, with the largest contribution to total emissions observed in the Stationary Combustion sector (C_OtherStacionaryComb) (Figure 7). The remaining sectors maintained their emissions during this period, except for the Solvents sector (E_Solvents) and the industry sector (B_Industry), with decreases of 90.8% and 23%, respectively.

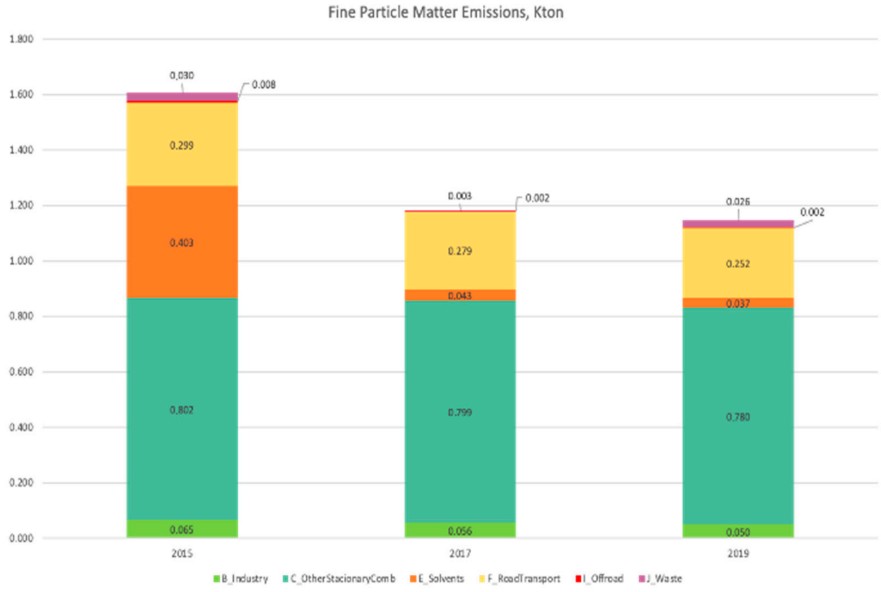

**Figure 7.** Fine particulate emissions of PM in the Municipality of Porto for 2015, 2017, and 2019, by sector. Source: APA.

### 3.2. Initiatives to Preserve and Enhance Cultural Heritage

The worsening of climate change is currently an undeniable reality. From periods of extreme drought to large forest fires, extreme climate events are expected to be increasingly frequent in the coming years, according to the European Environment Agency [13]. This situation necessitates a review of the forecast metrics used until now, which appear to be inappropriate and out of context for extreme weather events [14].

Table 1 illustrates future climate projections for the metropolitan area of Porto according to the Portuguese Environmental Agency.

This distribution of extreme climate events will not be uniform and they will not have the same magnitude throughout the urban context. However, it points out that municipalities with higher urban density will be the ones most affected by extreme climate change [14]. Therefore, it is essential to develop and implement a coherent set of measures to mitigate potential climatological impacts, safeguarding not only the population and the environment, but also cultural heritage [15].

In the European context, since 1992, with the Maastricht Treaty, efforts have been made to promote sustainable development; also, the approval of the Aalborg Charter at the 1st Conference of Sustainable Cities and Towns in 1994 in Denmark marked a key moment, as it implemented important action strategies aiming to protect environmental resources from the perspective of improving socio-economic conditions [16]. Nevertheless, it was not until 2006 that the Municipality of Porto signed the Aalborg Charter, together with 18 other regions in the North of Portugal and Galicia. Two years later, the city joined the initiative promoted by the European Commission and signed the "Covenant of Mayors" in 2009, committed to decreasing its $CO_2$ emissions by 45% by 2020, based on investments around renewable energy and efficient energy demand actions [15].

**Table 1.** Possible extreme weather events for the Porto metropolitan area. Diagram adapted from the report presented by the APA "Emissions of air pollutants by municipality—2015, 2017 and 2019". Source: AMP.

| Extreme Weather Event | Effects |
|---|---|
|  Extreme Temperatures | sea level rise; coastal erosion; saltwater intrusion into the groundwater table; storms; forest fires; deterioration of air quality; energy blackouts; an increase in the number of accidents; the development of new pathologies; increased morbidity and mortality |
|  Intense Rainfall | floods; landslides; groundwater pollution; interruption and damage to energy supply, transportation, communications and water supply; financial impacts on the built environment |
|  Drought | contamination of soils and subsurface and groundwater resources; reduced water quality; interruptions affecting the public water supply; increased cost of water; increased prevalence of certain illnesses |
|  Fast Winds | tree toppling; loss and damage to human life, equipment, infrastructure, among others |

According to the Municipal Strategy Plan for Adaptation to Climate Change of the municipality of Porto, the last emissions report showed compliance with at least half of the proposed target [15]. The implementation of Porto's Metro Integrated Business Strategy between 2007 and 2027 contributed to achieving this commitment, establishing as priorities decreases in emissions and energy intensity and increases in the efficiency of passenger and freight transportation [17]. The Porto Gravítica Project in 2007 defined a new water supply system in the municipality [18] and also the Operational Program Sustainability and Efficiency in the Use of Resources—PO SEUR, started in 2014, proposed to achieve sustainable growth based on a low-carbon economy [19].

In 2009, the Porto Metropolitan Area (AMP) joined the EU CO2 80/50 project together with fourteen other European metropolitan regions, committing to reduce greenhouse gas (GHG) emissions in the metropolitan areas by 80% before 2050 compared with 1990. In 2016, the Municipality of Porto became an integral part of the ClimAdaPT. Local Project—Municipal Strategies for Adapting to Climate Change, which was based on raising awareness for climate change in Portugal by developing municipal strategies to mitigate these factors [20].

In 2021, the Porto City Council became part of the Green City Accord, a voluntary movement on the part of European mayors, consisting of the implementation of several strategies in five areas of action: water quality, air quality, reduction in noise levels, biodiversity and nature, waste, and economy [21].

Recently, the Municipality of Porto launched the Porto Climate Pact in January 2022 towards a carbon-neutral municipality that is more sustainable and better prepared for climate extremes to achieve a neutral balance in emissions. This project endorsed the Green City Agreement and the Mayors' Pact for the Climate [22].

All these actions are extremely important as they help achieve the goals proposed by the 2030 Agenda Plan and its sustainable development goals (SDGs) [23].

## 4. Methodology

The methodology adopted was based on the anomalies survey through local photographic records and surroundings to assess the effects of pollution. This methodology was based on using various national and international documentary sources about the churches under study, highlighting the data provided by SIPA (Information System for Architectural Heritage) and DGPC (Directorate-General for Cultural Heritage). In addition to consulting the existing bibliography, a photographic record of each location and its surroundings was also created to survey existing anomalies. Finally, a summary table was created to characterise the surroundings of each case study. In the future, we will consider using photogrammetry as regularly as possible to understand the anomalies' evolutions.

Data provided by the APA were consulted to study the impact of pollution in the metropolitan area of Porto, and the information was subsequently gathered into graphs. The summary tables and pathology maps were based on the model referred to in the article "Adaptation of Damage Map for Historic Buildings With Pathological Problems: Case Study of the Carmo Church in Olinda Pernambuco" [24] and also "ICOMOS-ISCS: Illustrated Glossary on Stone Deterioration Patterns" [25] through a photographic record of the anomalies present in each building, highlighting the most intensely deteriorated areas, each one represented by different colours.

## 5. Case Studies: Carmelitas Church, Vitória Church, and São João Novo Church

### 5.1. Characterisation of the Surroundings

The deterioration of buildings occurs when damage to the constituent components or materials is caused by the action of degradation agents [26]. Facades, one of the most important systems in a building, depending on their position concerning the environment in which they are located, are the first to receive external aggression [27]. In general, the deterioration of building facades is directly related to a set of variables that particularly affect the plinths, windowsills, and walls, places that, depending on their orientation, will experience a lesser or higher incidence of sunlight and rainwater [28].

On the other hand, the existence of vegetation and/or other elements that offer protection to the façade will influence its exposure conditions, providing better durability and performance of its constituent materials. It should be noted that the proximity of traffic routes and car circulation, and the consequently higher release of atmospheric gases, are very damaging to building facades, contributing to stone decay (Table 2). Also, the proximity of the sea to these buildings could compromise them due to the atmospheric conditions and microorganisms present in these areas [29].

**Table 2.** Summary table of the surroundings of each studied church. Source: Author.

| Churches | Location | Main Façade Orientation | Traffic and Pedestrians | Surroundings with Buildings and Vegetation |
|---|---|---|---|---|
| **Igreja das Carmelitas** | City centre | South | Quite a lot | No |
| **Igreja de São João Novo** | Next to the river | North | Some | Yes |
| **Igreja da Vitória** | City centre | North | Some | Yes |

The image below locates and identifies each case study (Figure 8).

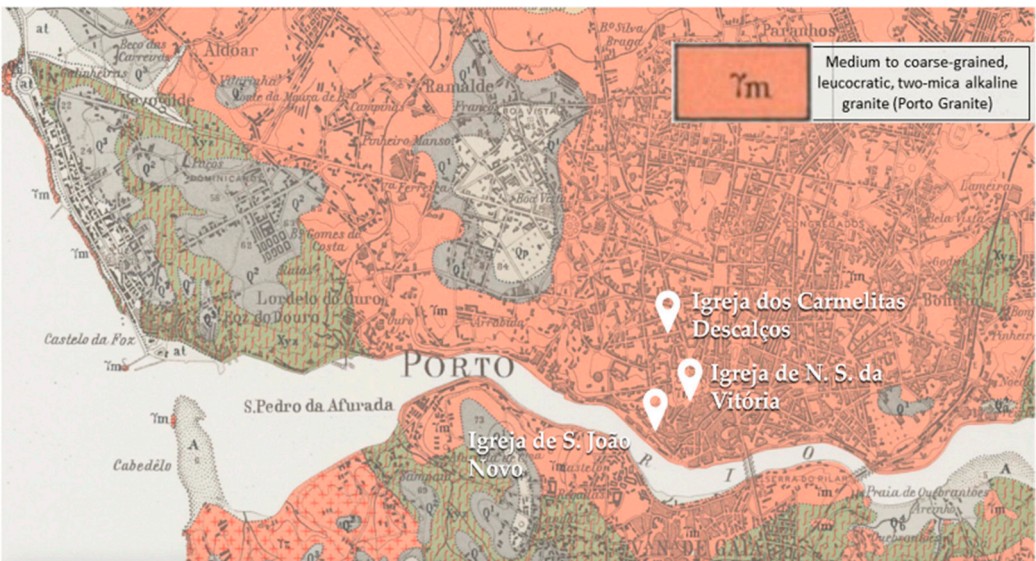

**Figure 8.** Locations of the churches under study on the map considering the geological setting of the study area in the Porto area (granite is the main source in terms of geological context—adapted from [30]).

Located in the heart of the historic centre of Porto, at the intersection between Carmo Street and Praça/Square Carlos Alberto (Figure 9), the surroundings of Carmelitas Church are characterised by being quite busy, given the number of tourists in this area, and also "unprotected" due to the lack of vegetation of buildings nearby, making it more susceptible to external agents, such as rainwater and the water–wind combination. Its main façade faces south, receiving a higher level of solar radiation during the year compared with the other façades facing other orientations [31].

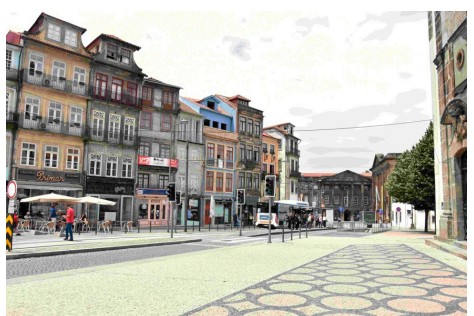
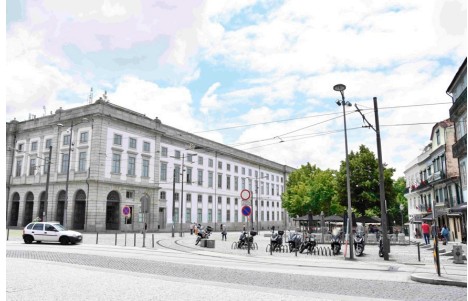

**Figure 9.** View of Carmo and Carlos Alberto Square, Porto. Source: Author.

The high level of car traffic in this area may explain the degradation of the granite stone and tiles that cover the facade. As there are no buildings or vegetation to protect the facade, it is more susceptible to external agents, such as rainwater and the water–wind combination.

Regarding Vitória Church, although located close to Carmelitas Church, the surrounding area presents less pedestrian movement and less car traffic. Located between São Miguel and São Bento da Vitória streets (Figures 10 and 11), its façade faces north, resulting in a cold façade that does not receive direct solar radiation during most of the year [31].

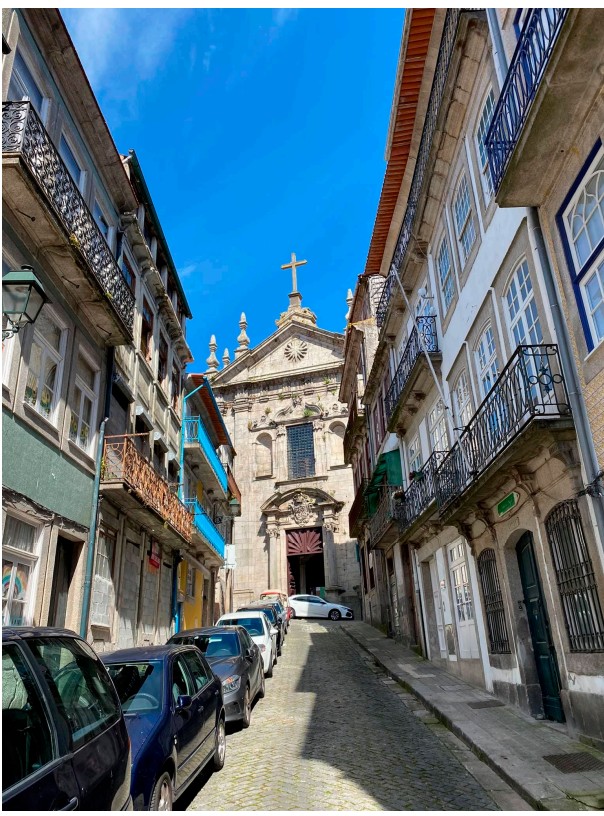

**Figure 10.** View of S. Miguel Street and Vitória Church. Source: Author.

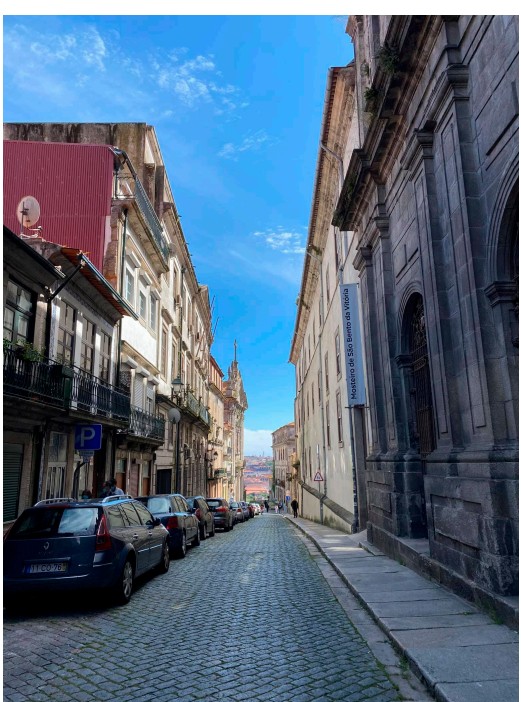

**Figure 11.** View of S. Bento da Vitória Street. Source: Author.

A little further down, we find São João Novo Church with its main façade facing north, towards Largo São João Novo (Figure 12). Similar to Vitória Church, this church also presents a cold façade, making it more susceptible to external agents; its surroundings have some car traffic and pedestrian movement.

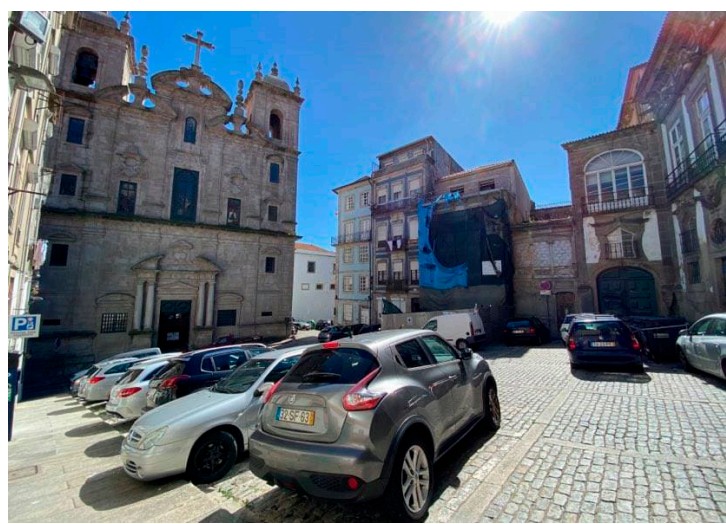

**Figure 12.** View of São João Novo Square. Source: Author.

*5.2. Carmelitas Church*

5.2.1. Characterisation and Historical Context

Originally known as the Order of "Irmãos da Bem-Aventurada Virgem Maria Monte Carmelo", or simply the Order of "Carmo" or "Carmelitas", this religious order emerged at the end of the 11th century on Mount Carmel, near the current city of Haifa in Israel. In the 16th century, resulting from the Order of Mount Carmel reform, a new branch, the "Discalced Carmelites", was guided by Saint Teresa of Ávila and São João da Cruz. Pope Clement VIII granted its autonomy in 1593. After the first convent was built by this order in Portugal in 1581, its rapid expansion led to the establishment of several convents across the country, reaching Porto in 1616 [32].

The construction work on Carmelitas Church began in 1619 and lasted for the first half of the 17th century, ending only in 1628 and its decorative campaign in 1650 [32].

A typical example of Baroque architecture (Figure 13), this church has a cruciform plan attached to a quadrangular bell tower on the west side of the main facade. This church has three floors covered in tiles and granite stone, with a pyramidal roof. We find a portal topped by a small oculus on the first floor. The main façade, on two levels and facing south, is topped with a triangular pediment consisting of a cross and three pinnacles flanked on each side. In the centre stands the coat of arms of the Discalced Carmelites Order, topped by the royal crown. On the first level, there are three rounds across, isolated by Doric–Roman pilasters, the central one larger than the rest. Above these stand the images of Saint Joseph, Saint Teresa, and Our Lady of Mount Carmel in the centre, with this figure topped by an interrupted pediment. Still on the same level, three large windows open, the side ones curved [32].

Inside are six chapels in the nave, three on each side, with altarpieces finished in gilded woodwork (Figure 14). The main chapel displays a gilded altarpiece from 1767 with 18th-century paintings. The sacristy has a coffered ceiling decorated with various paintings [32].

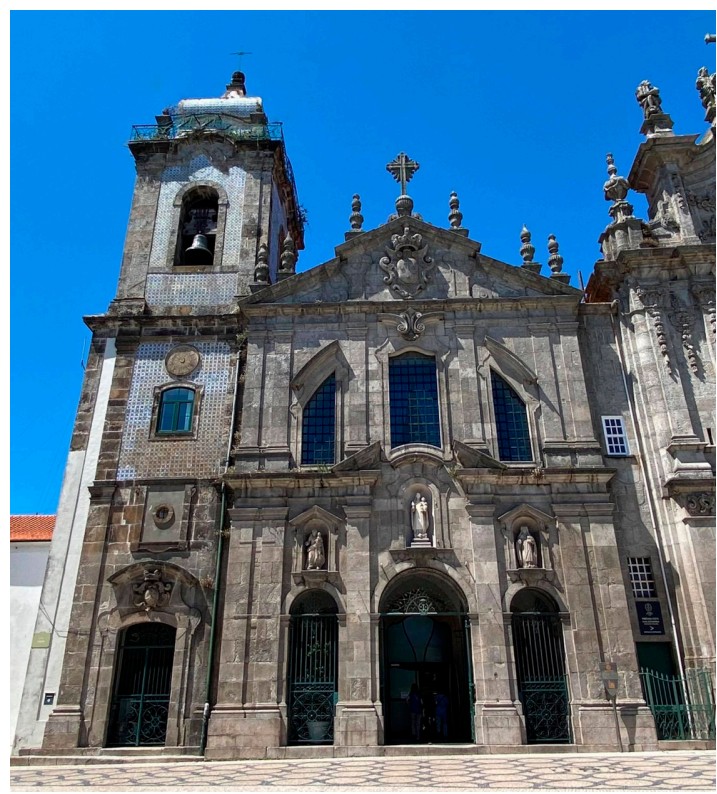

**Figure 13.** Carmelitas Descalços Church. Source: Author.

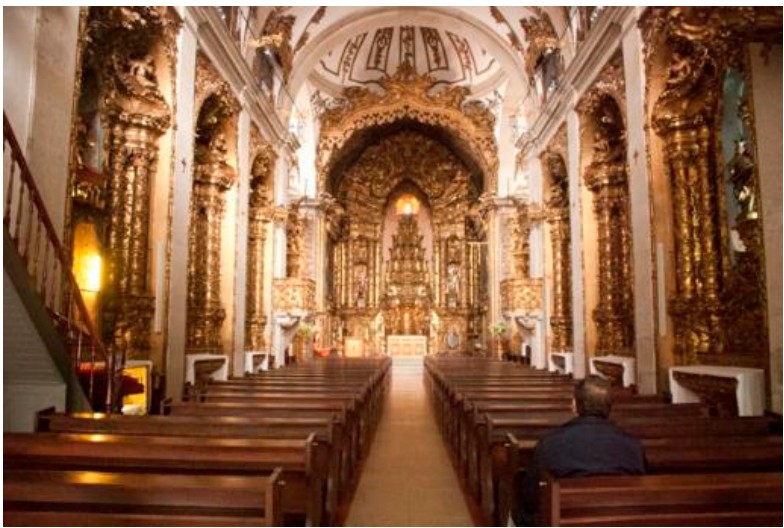

**Figure 14.** Interior of Carmelitas Descalços Church. Source: Author.

5.2.2. Damage Mechanisms

Based on the map model for identifying anomalies in the buildings referred to in the methodology [24,25], we prepared a summary table for each case study and a detailed map of the degradation presented on the façade. The execution was divided into two distinct phases: photographic documentation of the place of worship and its surroundings (Figures 15 and 16) and the creation of an anomalies map of the front facade of each church using image-editing software (Affinity Photo® version 1.10.1). Carrying out this type of mapping using a colour scheme allowed us to collect extensive information on the various types of degradation in each building.

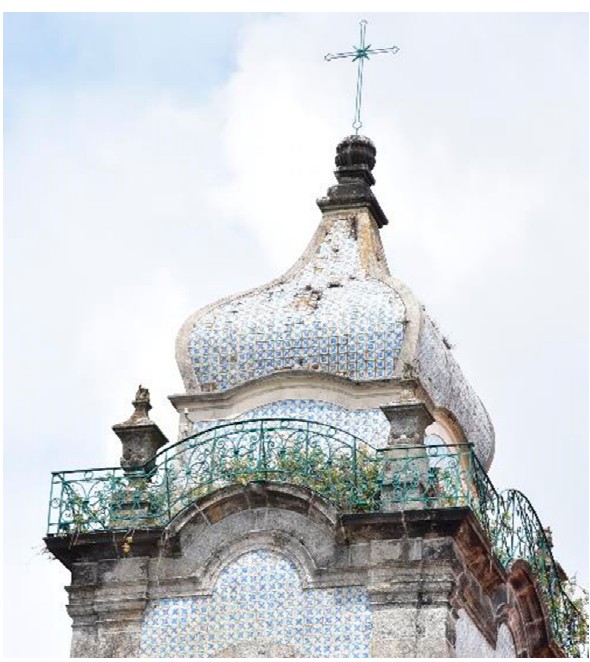

**Figure 15.** Details of the anomalies present on the south facade of Carmelitas Church. Source: author.

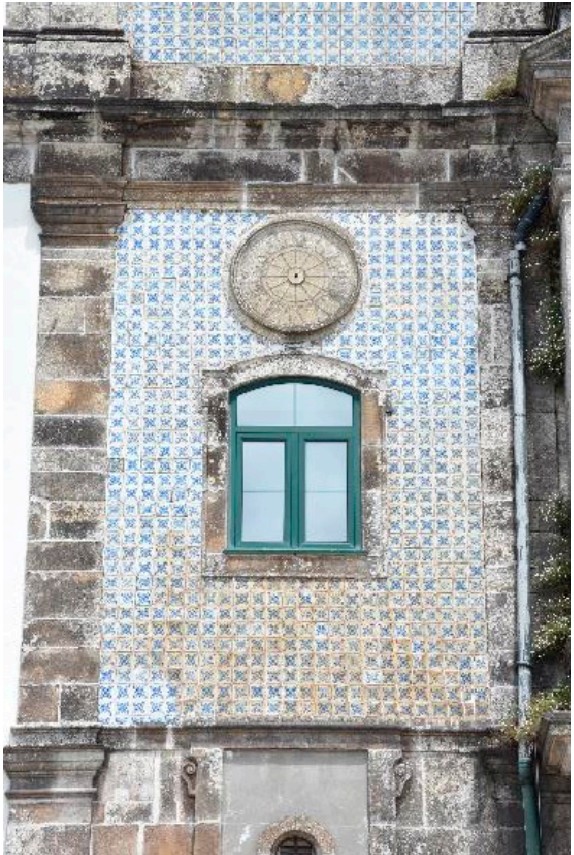

**Figure 16.** Details of the anomalies present on the south facade of Carmelitas Church. Source: author.

The anomalies mapping (Figure 17 and Table 3) revealed the existence of dirt and black crusts on the main facade, the presence of vegetation (with a greater incidence in the bell tower), humidity, cracking, and detachment of the glaze on the tiles, and, finally, chromatic changes, scaling, granular disintegration, and biological colonisation in stone masonry.

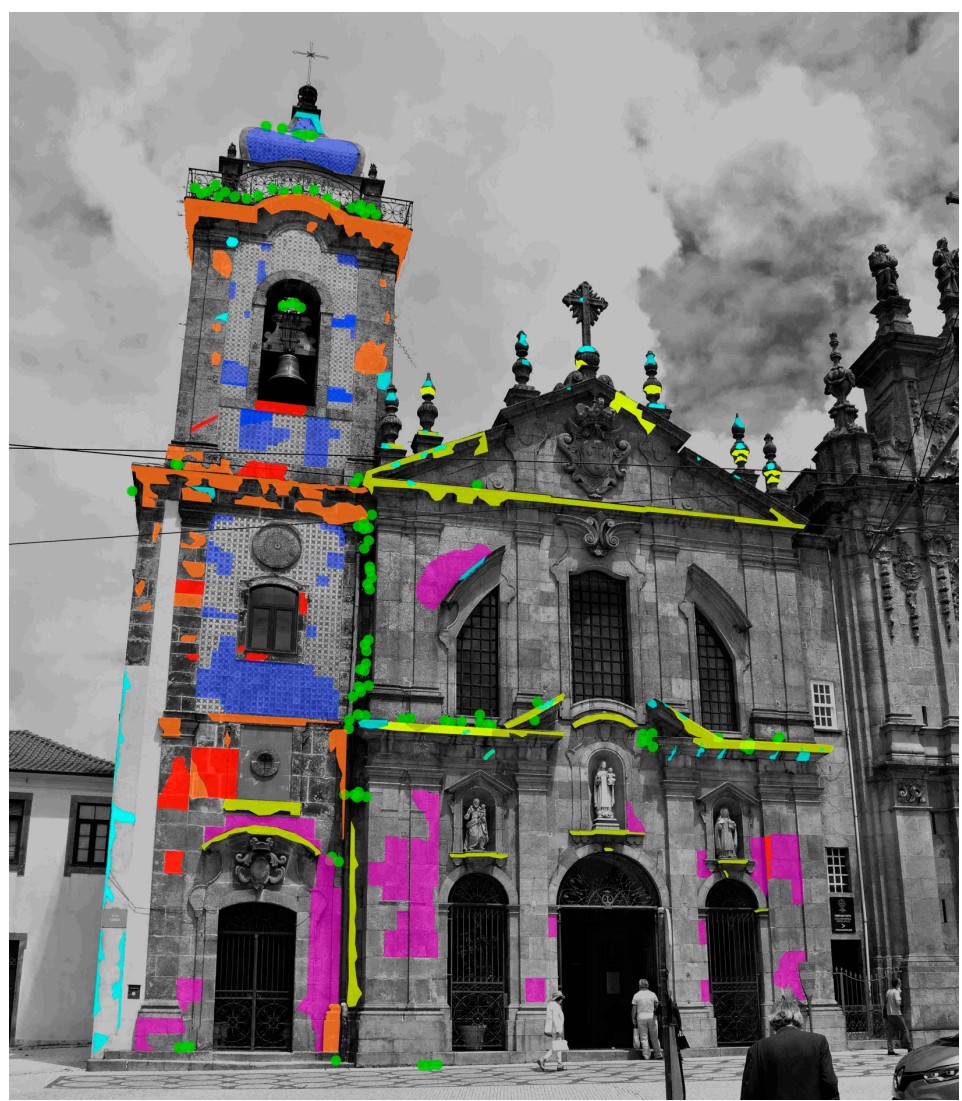

**Figure 17.** Anomalies map of the front facade of Carmelite Church. Source: Author.

**Table 3.** Summary table with the pathologies identified on the facade of Carmelite Church (according to [25]). Source: Author.

| Type of Deterioration | Occurrence | Symbolism |
|---|---|---|
| **Biological Colonization** | X | 🟩 |
| **Crack** | X | 🟪 |
| **Crust** | X | 🟧 |
| **Detachment of plaster** | X | 🟦 |
| **Discolouration** | X | 🟥 |

**Table 3.** *Cont.*

| Type of Deterioration | Occurrence | Symbolism |
|:---|:---:|:---:|
| **Greenery** | X | 🟩 |
| **Humidity** | X | 🟨 |
| **Scaling** | X | 🟪 |

*5.3. São João Novo Church*

5.3.1. Characterisation and Historical Context

Built on the hillside that meets the Douro River, São João Novo Church shows us all its magnificence over the river and the historic centre that developed on the north bank. According to Silva [33], this building was built between the 16th and 18th centuries on top of a primitive hermitage dedicated to the Saint John the Baptist patron. Its brief existence was as the headquarters of the parish of São João de Belmonte, which, shortly after its extinction in 1604 by Bishop D. Frei Gonçalo de Morais, was divided between the Churches of São Nicolau and Nossa Senhora da Vitória, and subsequently donated to the Ermitas Calçados de Santo Agostinho to mark their presence in the city of Porto. Since it did not meet the needs of the Order, it was demolished, and São João Novo Church was built [34].

The construction process of the new church was long due to the low monetary resources, lasting for more than one century, and it was only completed around 1779, which led to changes in the building's plan [33].

The church's facade faces north, as does the monastery, oriented to Praça and São João Novo Street. On the west side, there would eventually be the Priest's house, the chapel of Nossa Senhora da Esperança, and the conventual space facing south [34]. This building has a Latin cross plan consisting of a nave, with on each side two interconnecting chapels flanked by granite vaults, a transept with short arms, and, on the sides and at the top, baroque altarpieces. The main chapel boasts a baroque carved altarpiece, topped by a vault [33].

The main facade has three floors (Figure 18), the first two open and the third closed, built with granite masonry. According to Santos [34], although the entire facade presents a good construction, there is a discontinuity in the constructive language, with some inaccuracies in the proportions located between the columns and pedestals on the main facade. The church's interior is of inferior construction [33], with a Roman Neoclassical style.

The construction materials used were granite masonry on the church's exterior and plaster on the convent walls [34]. The walls of the main chapel (Figure 19), the side chapels, and the nave are in pink marble, and the sacristy and cloister are rendered with tiled panelling [33].

According to Sereno and Noé [32], this church underwent three interventions. In 1995, the sacristy and chancel were conserved and restored, and the electrical installation was installed; in 1996, the roofs and exterior walls were repaired, the tile panels covering the Chapel of Santa Rita were recovered, and the roof of the parish house was renovated; and finally, in 1998, all the roofs inside the church were rebuilt [33].

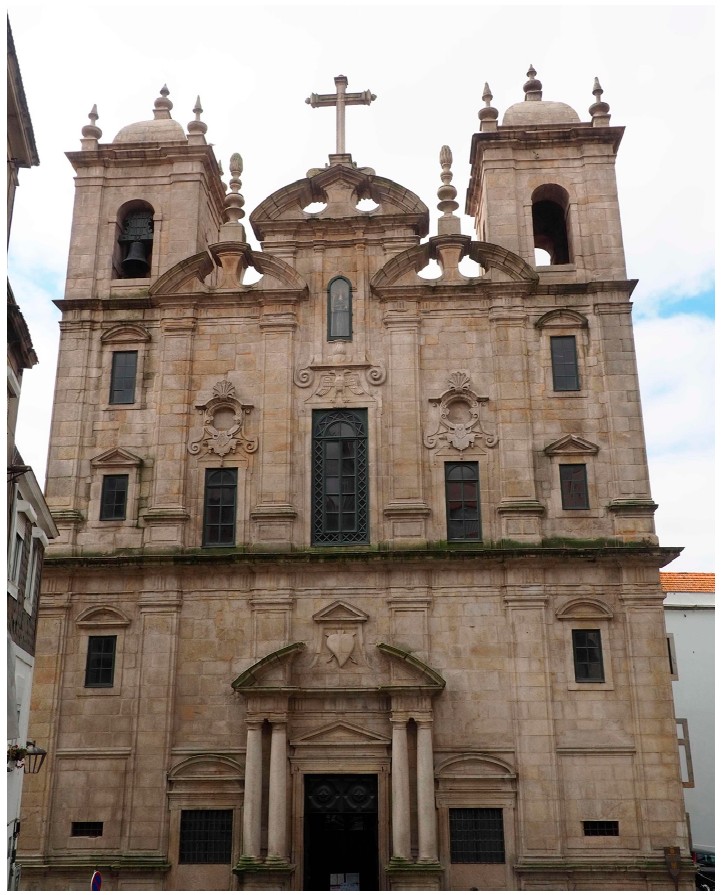

**Figure 18.** Main façade of the Church of St. São João Novo. Source: Author.

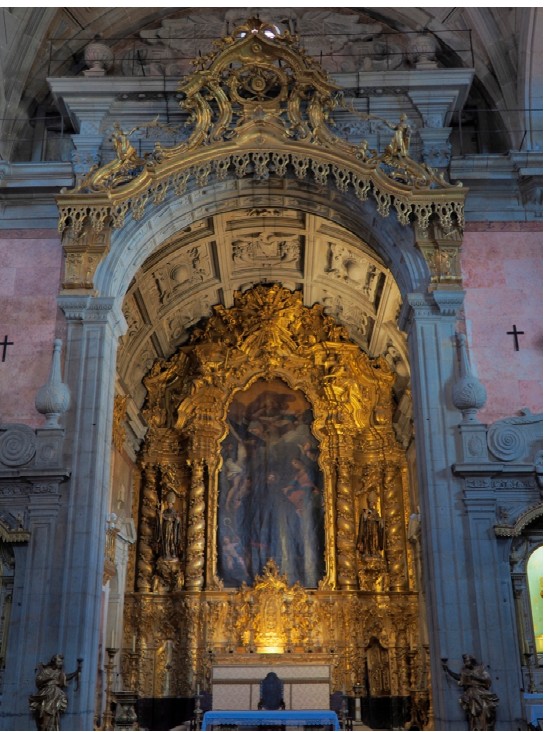

**Figure 19.** Main chapel of the Church of São João Novo. Source: Author.

5.3.2. Damage Process

After completing all the photographic documentation (Figures 20–22) and creating the degradation map (Figure 23 and Table 4), we highlight the main pathologies on the main facade of São João Novo Church: biological colonisation, cracks, and humidity, with a higher incidence on the pediments and columns, as well as some lacunae across the entire facade, open joints, and vegetation, with greater predominance in the upper part of the building.

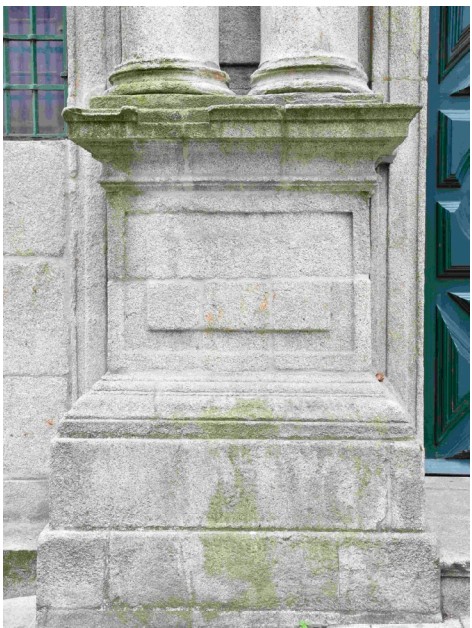

**Figure 20.** Identification of the various pathologies on the north facade of São João Novo Church. Source: Author.

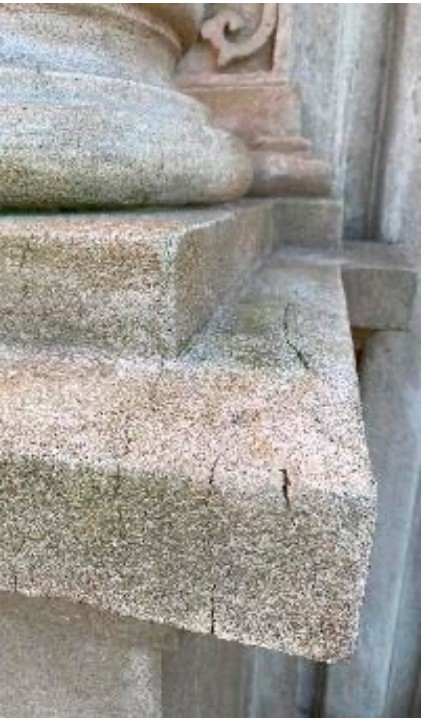

**Figure 21.** Identification of the various pathologies on the north facade of São João Novo Church. Source: Author.

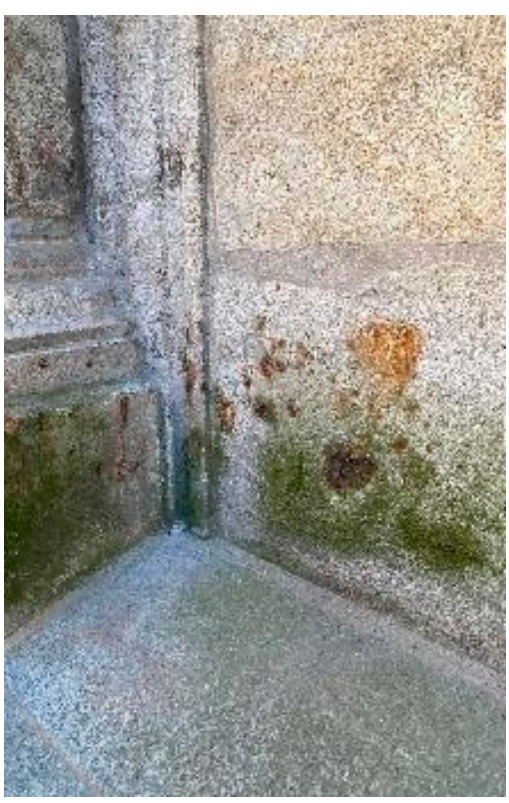

**Figure 22.** Identification of the various pathologies on the north facade of São João Novo Church. Source: Author.

**Table 4.** Summary table with the pathologies identified in the Church of São João Novo (according to [25]). Source: Author.

| Type of Deterioration | Occurrence | Symbolism |
|---|---|---|
| **Biological Colonization** | X | |
| **Crack** | X | |
| **Greenery** | X | |
| **Humidity** | X | |
| **Missing Part** | X | |
| **Open Joints** | X | |

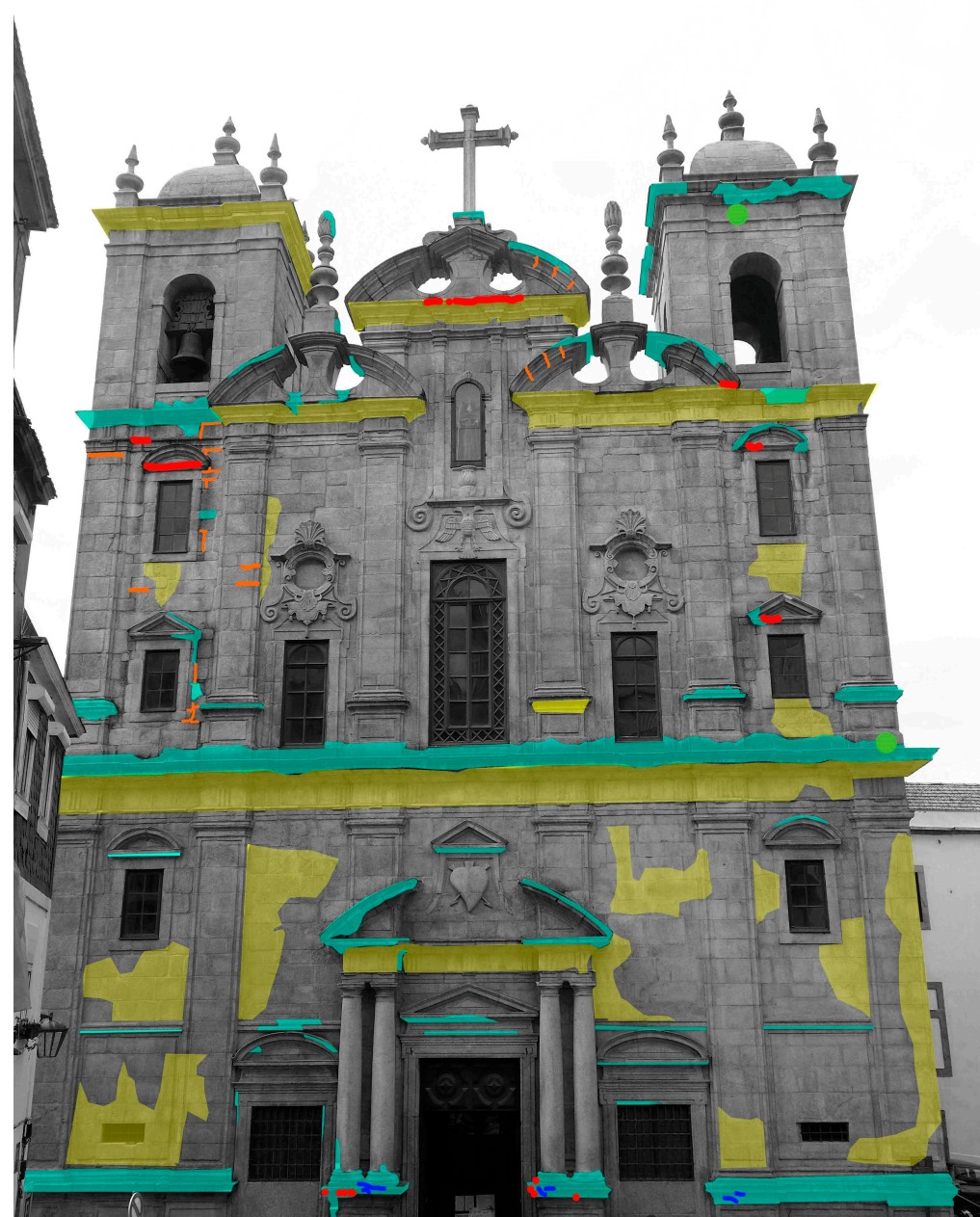

**Figure 23.** Anomalies map on the front facade of Carmelite Church. Source: Author.

*5.4. Vitória Church*

5.4.1. Characterisation and Historical Context

The primitive Church of Nossa Senhora da Vitória (Figure 24), located in the old part of Porto City, in the parish of Vitória, was located within the defensive walls next to Praça do Olival, occupying part of the Judiaria Nova, where Jews were once concentrated by order of D. João I. This church was built in 1539 under the authority of Bishop D. Frei Marcos de Lisboa [35].

In 1755, given its advanced state of degradation, its rehabilitation and rebuilding began under the leadership of Bishop D. Frei António de Sousa using alms from the faithful. It was inaugurated on 5 August of 1769, during the period of vacancy of the Cathedral (1766–1770) [36]. Therefore, the church's parish services were temporarily transferred to the Chapel of S. José das Taipas [37].

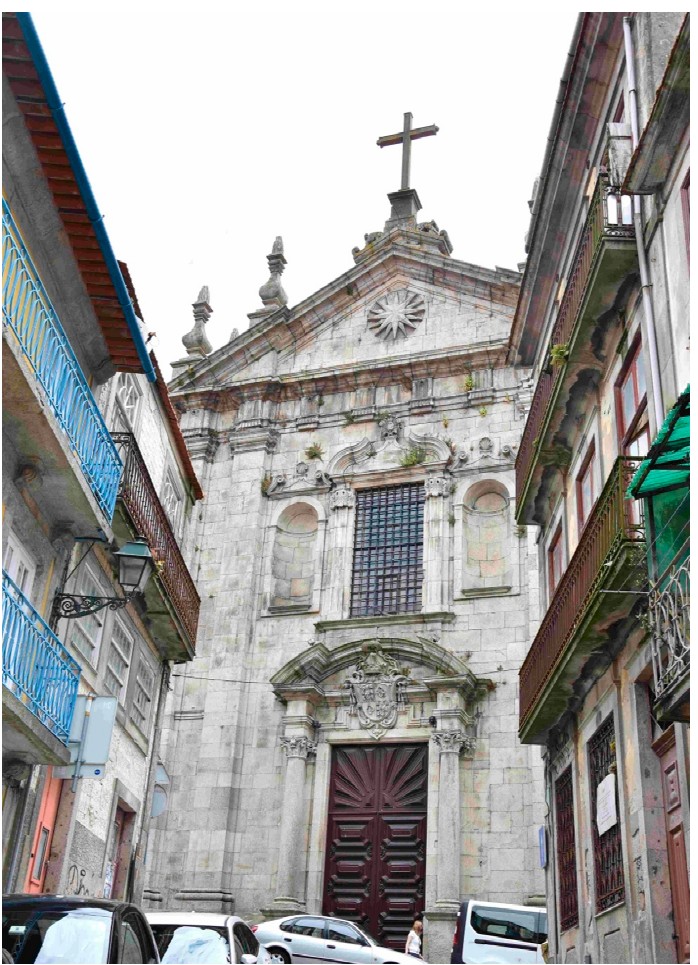

**Figure 24.** Main facade of Nossa Senhora da Vitória Church, Porto. Source: Author.

During the period of the Siege of Porto in the middle of the 19th century (1831–1834), the church was damaged when hit by the artillery of the Miguelistas installed on the opposite side of the Douro River, once again forcing the transfer of parish services to the neighbouring convent church of Porto. After the Siege ended, it was only in 1852 that the restoration works at Vitória Church were completed. In 1874, a fire destroyed the main altar and, consequently, the church's image, which led to some interventions inside the church [38].

Built under the influence of the classical style, Vitória Church is based on a possible Synagogue due to the strong Jewish presence between São Bento da Vitória Street and Bataria da Vitória Street, a toponym that takes us back to the battle during the Siege of Porto mentioned above [37]. Currently, it is possible to see a bullet inlaid in the south wall, a testimony to the liberal struggles (Figure 25) [36].

This church has a west-facing main façade. The sacristy and parish residence flank the opposite façade. It consists of a single nave covered in stucco covered by a brick vault. It has four side chapels and a chancel accompanied by a groin vault and a large pelmet, and two pulpits topped by rococo carved altars, topped by Neoclassical pelmets (Figure 26) [39]. A quadrangular bell tower is connected to this last chapel, with stonework walls decorated with fires and a granite pyramidal dome featuring a lunette on each face, topped with a metal cross resting on a sphere [35].

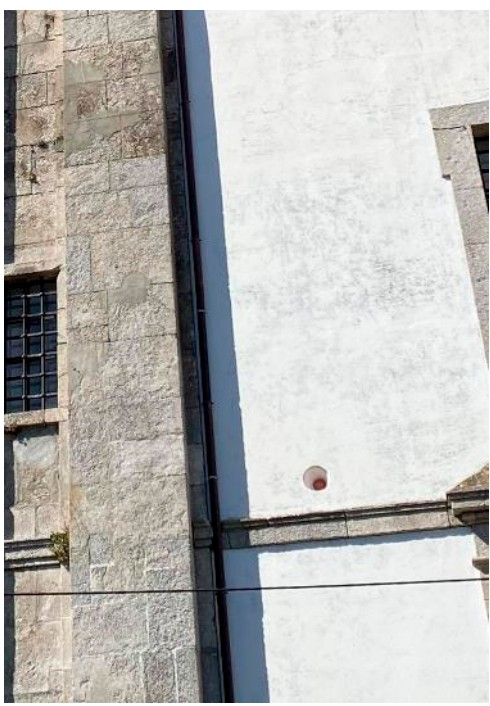

**Figure 25.** Bullet inlaid in the south wall. Source: Author.

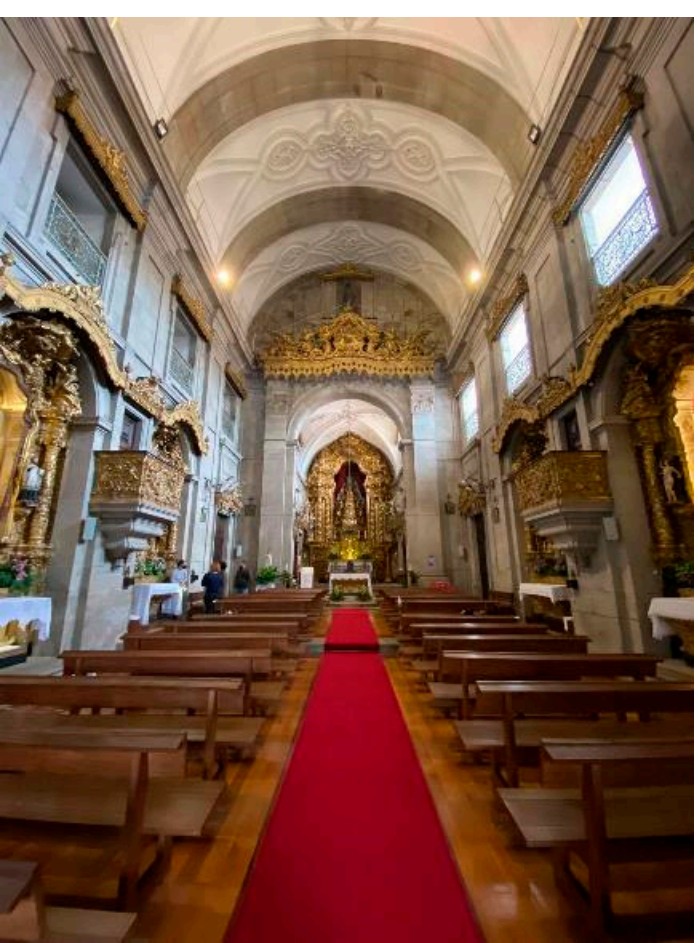

**Figure 26.** Interior of Nossa Senhora da Vitoria Church. Source: Author.

5.4.2. Damage Process

After detailed analysis (Figures 27–30) and mapping of anomalies (Figure 31 and Table 5), the main anomalies identified on the main façade of Vitória Church were vegetation, biological colonisation, gaps, black crusts, dampness, cracks, and platelets.

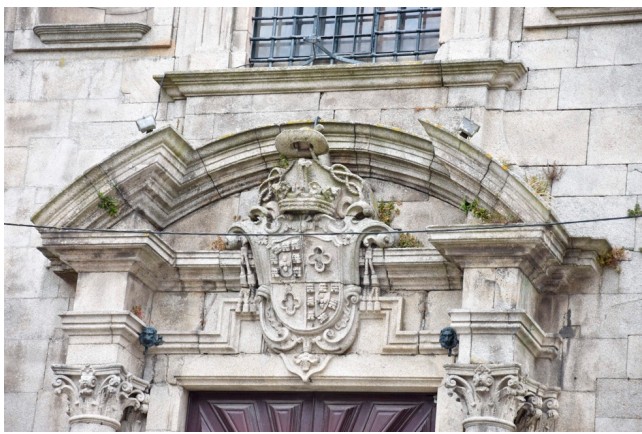

**Figure 27.** Identification of the various pathologies on the north façade of Vitória Church, Porto. Source: Author.

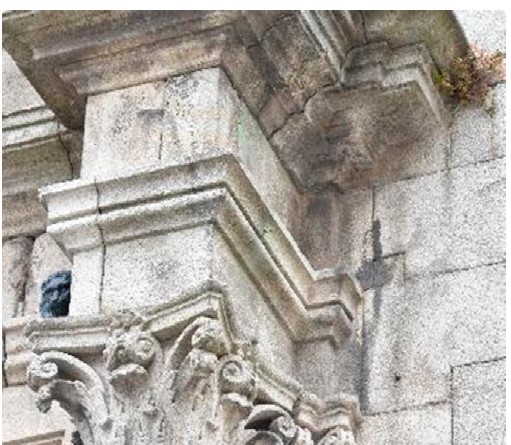

**Figure 28.** Identification of the various pathologies on the north façade of Vitória Church, Porto. Source: Author.

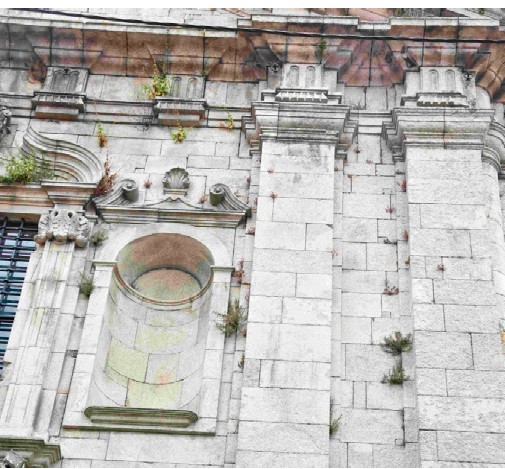

**Figure 29.** Identification of the various pathologies on the north façade of Vitória Church, Porto. Source: Author.

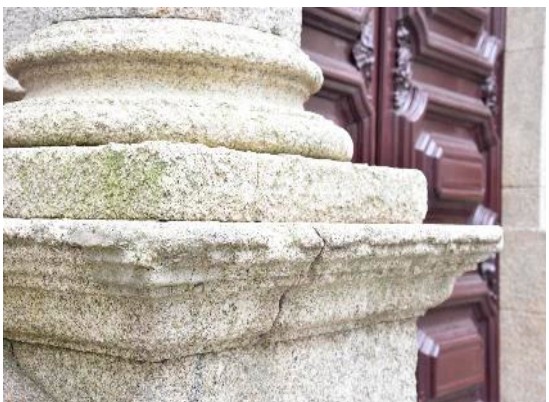

**Figure 30.** Identification of the various pathologies on the north façade of Vitória Church, Porto. Source: Author.

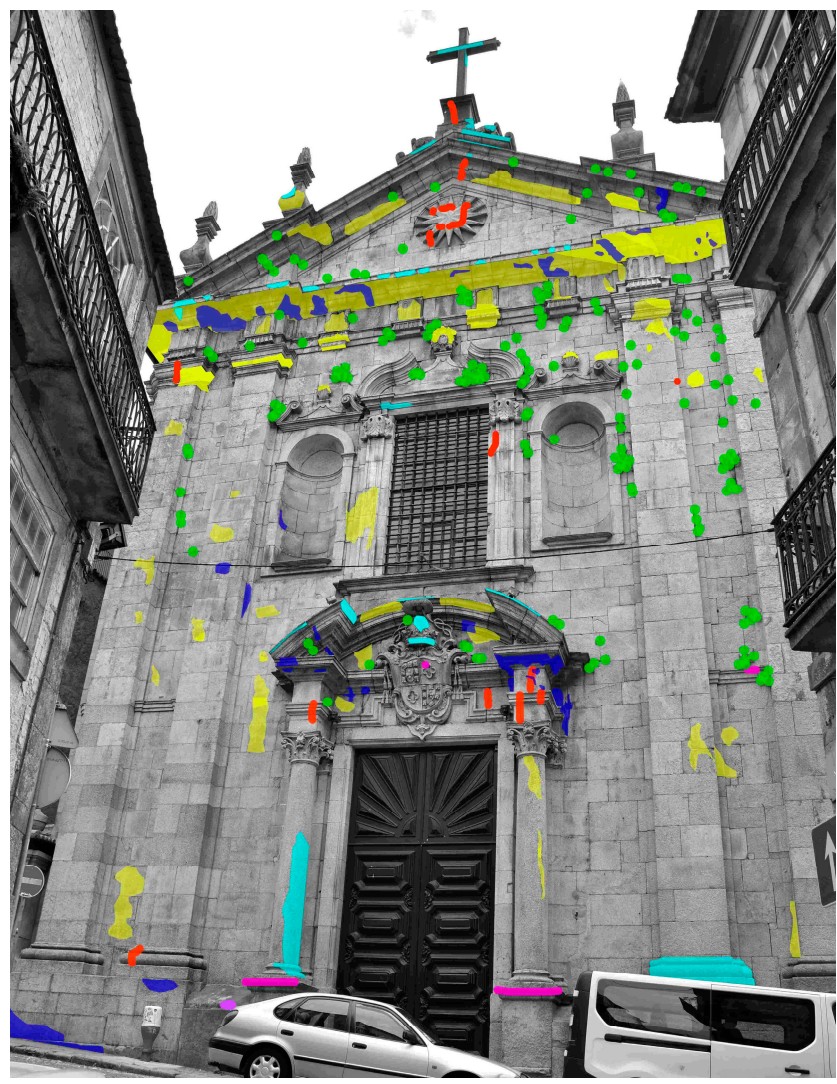

**Figure 31.** Anomalies map on the front facade of Vitória Church. Source: Author.

**Table 5.** Summary table of pathologies identified on Vitória Church (according to [24]). Source: Author.

| Type of Deterioration | Occurrence | Symbolism |
|---|:---:|:---:|
| **Biological Colonization** | X | |
| **Crack** | X | |
| **Crust** | X | |
| **Fragmatation** | X | |
| **Greenery** | X | |
| **Humidity** | X | |
| **Missing Part** | X | |

## 6. Discussion of Results and Conclusions

The intensification of climate change in recent decades has led to the early deterioration of cultural heritage. Municipal and inter-municipal projects and cooperation have been proven to be an asset in combating these factors.

In Porto, since 2006, with the Aalborg Charter, it has been possible to see a significant improvement in climate change mitigation, but there is still a long way to go.

This study revealed the most prevalent types of degradation present at each study site. We observed that biological colonisation, greenery, cracks, humidity, and gaps were the most frequent pathologies in the study cases (Table 6). All of these pathologies are in some way related not only to Porto's climate, characterised by long periods of precipitation and high relative humidity, but also to its main façade's orientation. Facades facing north, such as São João Novo Church and Vitoria Church, are naturally more prone to humidity and biological colonisation, as they are naturally colder facades.

We could also clearly see the direct consequences of the surroundings on worship places. Of all the churches studied, Carmelitas Church had a rather "unprotected" environment, characterised by its large influx of tourists and consequently a large amount of car traffic, which directly affected the façade, such as chromatic change, detachment of plaster, and scaling.

Some difficulty was observed in identifying all the anomalies present on the facades of the studied churches due to the viewing angle from below concerning the grandeur of these buildings and uncertainty when identifying. Therefore, we plan to fill this gap in future work by creating even more specific and correct anomaly maps for each case study and also conducting ageing tests on some building materials. These ageing tests consist of the simulation of natural weathering by reproducing some climatic events (rain, wind, air pollutants, temperature, relative humidity, etc.) [40].

In the future, we intend also to use the photogrammetry technique at the facades, which will allow us to carry out a survey with much more detail and update our knowledge regarding the degradation problems and their evolution over time. This technique could be used several times in the future to monitor the evolution of the anomalies. Although we will also consider the methodology used in the present research, photographs will be used to compare them in other surveys with the ones in the present research and the future, and

they will be taken from several angles, considering the facade and its details. Using a laser scanning monitoring technique would also be helpful, although it depends on the funding we can achieve.

**Table 6.** Summary table of pathologies identified in all the worship places studied (according to [25]). Source: Author.

| Type of Deterioration | Carmelitas Church | São João Novo Church | Vitória Church |
|---|---|---|---|
| **Biological Colonization** | X | X | X |
| **Crack** | X | X | X |
| **Crust** | X | | X |
| **Detachment of plaster** | X | | |
| **Discolouration** | X | | |
| **Fragmatation** | | | X |
| **Greenery** | X | X | X |
| **Humidity** | X | X | X |
| **Missing Part** | | X | X |
| **Scaling** | X | | |
| **Open Joints** | | X | |

Climatic variation introduces additional challenges and considerations for conservation efforts. Flexibility, adaptability, and a proactive approach to address emerging issues are crucial for successful conservation. Thus, we can propose some suggestions to understand how climatic variation can affect or modify the conservation of this religious architectural heritage.

Changes in precipitation patterns, including more intense and frequent rainfall, can increase moisture levels. This excess moisture can lead to problems such as water infiltration, dampness, and mould growth.

Climate changes can influence the distribution and behaviour of pests and organisms representing a risk to cultural heritage. For example, warmer temperatures may increase insect activity, potentially accelerating the degradation of organic materials. On the contrary, drought periods can affect heritage stability [41].

The impacts of climate change can influence local communities and their relationships with cultural heritage, such as changes in traditional practices and economic activities that may affect the stewardship and care of their heritage.

Climate-related events can strain the resources available for cultural heritage conservation. Emergency response efforts, recovery measures, and long-term maintenance may require additional funding and attention, additional resources, and the development of planned conservation projects.

Conservation practices may need to be adapted to address new challenges arising from climate change. Increased costs and efforts are required for adaptive conservation measures in order to achieve the sustainable development goals of the United Nations, namely the 13th goal: Climate Action by the Year 2030 [42].

Addressing these challenges requires a multidisciplinary approach incorporating climate science, architecture, and conservation practices to ensure the long-term sustainability of built architectural heritage to face climate change.

Cultural heritage conservation practices should incorporate climate resilience strategies to address these challenges. This may include developing climate-specific conservation plans using sustainable materials and methods, using climate-resistant materials, and improving community engagement and awareness to ensure the continued protection of cultural heritage concerning changing climatic conditions. Collaboration among heritage

conservation professionals, scientists, policymakers, and local communities is crucial to developing effective and sustainable solutions [41].

**Author Contributions:** All authors contributed equally to the construction of this article. Conceptualization, F.M.S., M.A. and A.F.; Methodology, F.M.S., M.A. and A.F.; Investigation, F.M.S., M.A. and A.F.; Writing—original draft, F.M.S., M.A. and A.F.; Writing—review & editing, F.M.S., M.A. and A.F. All authors have read and agreed to the published version of the manuscript.

**Funding:** This research was supported by the GeoBioSciences, GeoTechnologies, and GeoEngineering Research Centre (GeoBioTec), Ref. UIDB/04035/2020, funded by FCT and FEDER funds through the Operational Program Competitiveness Factors COMPETE and by national funds (OE) through the FCT in the scope of the framework contract foreseen in numbers 4, 5, and 6 of article 23 of the Decree-Law 57/2016 of 29 August, changed by Law 57/2017 of 19 July.

**Data Availability Statement:** The original contributions presented in the study are included in the article.

**Conflicts of Interest:** The authors declare no conflicts of interest.

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
