# Peer review of "The Impact of Pollution on Cultural Heritage in the Historic Centre of Porto, Portugal"

_urbansci, doi:10.3390/urbansci8020031_

Round 1

Reviewer 1 Report

Comments and Suggestions for Authors

The paper “The impact of pollution on cultural heritage in the Historic  Centre of Porto, Portugal” is  interesting and well structured, but there are somethings to improve. In detail:

·         pag 2 line 55: why the centre has lost his attractive and polarising power?

·         In the Figure 1 is not very clear in my opinion, it is too small and the defensive wall is not well visible. The caption must have more description.

·         Chapter 3.1 line 93-97: you repeat the same things with the same words you used in lines 36-39 of the introduction, please change them.

·         Figure 2: change on the dx Metrological in Meteorological

·         In the chapter 4 Methodology, please explain the acronyms SIPA and DGPC

·         Table 2 is before the citation in the text I suggest to put it below the citation

·         Page 10 line 288-289 express the same concept that is below in the lines  283-284

·         I was in difficulty seeing the description of the surroundings and then the description of the churches, object of the study. I would prefer the opposite: first the churches and then I would have contextualized them in the environment surroundings.

·         ​I suggest (see Figures 13-21-29) to put larger photos of the churches and leave out the interiors that are not the subject of study, or at least briefly talk about them

·         Figure 15,16,17,19, 19 are not very beautiful, the phenomenology of degradation is not clearly visible

·         Why don't you refer to the UNI 11182 of 2006 or to the ICOMOS  paper on  nomenclature of degradation phenomena? I suggest doing so.

·         What are the sources from which to obtain information on the lithologies constituting the churches? It would be important to talk more about them in addition to mentioning as references.

·         Figures 23,24,25, 26,27 are not very beautiful, the phenomenology of degradation is not clearly visible

·         Figures 32,33,34,35 are not very beautiful, the phenomenology of degradation is not clearly visible

·         Increase the references on degradation problems I suggest some papers on this topic:

https://doi.org/10.1016/j.conbuildmat.2021.124311

https://www.mdpi.com/2073-4433/11/8/788

https://link.springer.com/article/10.1007/s12665-012-2161-6

Author Response

Author's Reply to the Review Report - Reviewer 1

We appreciate the reviewer's suggestions and opinions, which, of course, contribute to improving our article, so we have considered them. We thank the reviewer for the time and effort.

The reviewer was kind enough to make the comments and suggestions in great detail and some about the images. Given the difficulty of mentioning each of them here, we have chosen to make some of the changes directly in the text.

It can be noted that all comments and suggestions have been considered, contributing to an improvement of the article.

The paper “The impact of pollution on cultural heritage in the Historic  Centre of Porto, Portugal” is  interesting and well structured,

Thank you very much for your comments encouraging us to continue our investigations. We have considered your opinion and have already inserted a few more elements.

  • pag 2 line 55: why the centre has lost his attractive and polarising power?

Thank you. We completed the idea because it was incorrectly applied to the historic centre of Porto.

  • In the Figure 1 is not very clear in my opinion, it is too small and the defensive wall is not well visible. The caption must have more description.

Thank you very much for the suggestion. We have increased the size of the figure and improved the description placed in the legend.

  • Chapter 3.1 line 93-97: you repeat the same things with the same words you used in lines 36-39 of the introduction, please change them.

Thank you. We have deleted the redundant information.

  • Figure 2: change on the dx Metrological in Meteorological

Thank you very much for the suggestion. The change has been made.

  • In the chapter 4 Methodology, please explain the acronyms SIPA and DGPC

Thank you very much for the suggestion. We unfold the acronym in the text of the article and put here the information about these entities.

SIPA - Information System for Architectural Heritage is an information and documentation system on architectural, urban and landscape heritage Portuguese and of Portuguese origin or matrix managed by the Directorate-General for Cultural Heritage (DGPC).

DGPC - The Directorate-General for Cultural Heritage (DGPC) is responsible for the management of cultural heritage in mainland Portugal. An extended team, covering practically all technical and scientific domains and functionally structured in central services, based in Lisbon, and in Museums, Monuments and Palaces, located in different parts of the country, ensures a wide range of functions and provides a wide range of services.

  • Table 2 is before the citation in the text I suggest to put it below the citation

Thank you very much for the suggestion. Chart 2 was placed after the citation in the text.

  • Page 10 line 288-289 express the same concept that is below in the lines 283-284

Thank you very much, the repetition the duplication of the concept was removed.

        I was in difficulty seeing the description of the surroundings and then the description of the churches, object of the study. I would prefer the opposite: first the churches and then I would have contextualized them in the environment surroundings.

Thank you very much for the suggestion. However, given that the description of the surrounding area is done together, given the proximity of the three churches in the historic center of Porto, and the description of each one, as well as the characterization of the degradation mechanisms, is done separately, it would significantly change the structure of the article, which seems complex and unnecessary to us.

  • I suggest (see Figures 13-21-29) to put larger photos of the churches and leave out the interiors that are not the subject of study, or at least briefly talk about them

Thank you very much for the suggestion that has been taken into account.

  • Figure 15,16,17,19, 19 are not very beautiful, the phenomenology of degradation is not clearly visible

Thank you very much for the suggestion that has been taken into account. We've improved some photos and changed others.

  • Why don’t you refer to the UNI 11182 of 2006 or to the ICOMOS paper on  nomenclature of degradation phenomena? I suggest doing so.

Thank you very much for your suggestion. The nomenclature was adapted using as reference ICOMOS-ISCS : Illustrated glossary on stone deterioration patterns (this reference was added to the bibliography.

  • What are the sources from which to obtain information on the lithologies constituting the churches? It would be important to talk more about them in addition to mentioning as references.

Thank you very much for your suggestion. The figure 8 was substituted by a new one (Figure 8) considering the geological context.

  • Figures 23,24,25, 26,27 are not very beautiful, the phenomenology of degradation is not clearly visible

Thank you very much for the suggestion that has been taken into account. We've improved some photos and changed others.

  • Figures 32,33,34,35 are not very beautiful, the phenomenology of degradation is not clearly visible

Thank you very much for the suggestion that has been taken into account. We've improved some photos and changed others.

  • Increase the references on degradation problems I suggest some papers on this topic:

https://doi.org/10.1016/j.conbuildmat.2021.124311

https://www.mdpi.com/2073-4433/11/8/788 JÁ CITEI ESTE

https://link.springer.com/article/10.1007/s12665-012-2161-6

Thank you very much for suggesting these articles that have been taken into account and cited in the text and the bibliography.

Reviewer 2 Report

Comments and Suggestions for Authors

It is considered that the part of the contribution, in particular the description of the interior of the churches, is not relevant to the proposed topic and should be removed as it takes up space and does not provide useful information

The tables with the types of alterations include phenomena that are not present and become too long, only the alterations shown on the map should be identified 

It would be useful to make a table comparing the three monuments in a single representation, this would allow a better understanding of the differences found or similarities

The terminology used to describe alterations is not appropriate and does not refer to either the ICOMOS documents or the European CEN standards (EN 15898 and CEN/TS 17135)

The authors do not explain the process and segmentation methods used to define areas on surfaces. This topic would be of greater interest if it were explored in more depth and defined in terms of its limitations and advantages. Mappings expressed without a reference to parameters or selection functions do not give value to the contribution

No indication is given of the limits due to the definition of the images and, therefore, the accuracy of the measurements

No emphasis is placed on the need to have images taken orthogonally, perhaps with the use of a drone

The sentences in lines 36 and 93 are identical and therefore useless and repetitive at least one should be removed

The presence of pollution does not necessarily lead to conservation problems, as this depends on the environment, as also mentioned. There is a lack of a proposal for the possibility of benchmarks to understand how climate variation may affect or modify the conservation problems encountered

Author Response

Author's Reply to the Review Report - Reviewer 2

 We appreciate the reviewer's suggestions and opinions, which, of course, contribute to improving our article, so we have considered them. We thank the reviewer for the time and effort.

The reviewer was kind enough to make the comments and suggestions in great detail. Given the difficulty of mentioning each of them here, we have chosen to make some of the changes directly in the text.

It can be noted that all comments and suggestions have been considered, contributing to an improvement of the article.

Thank you very much.

It is considered that the part of the contribution, in particular the description of the interior of the churches, is not relevant to the proposed topic and should be removed as it takes up space and does not provide useful information

Thank you very much for the suggestion that has been considered, and the description of the interior of the churches has been substantially reduced.

The tables with the types of alterations include phenomena that are not present and become too long, only the alterations shown on the map should be identified

It would be useful to make a table comparing the three monuments in a single representation, this would allow a better understanding of the differences found or similarities.

Thank you very much for the suggestion. All tables have changed. Relatively a table comparing the three monuments in a single representation we consider that this table already exists in Figure 6. Even this has been changed.

The terminology used to describe alterations is not appropriate and does not refer to either the ICOMOS documents or the European CEN standards (EN 15898 and CEN/TS 17135)

Thank you very much for the suggestion that has been taken into account.

The authors do not explain the process and segmentation methods used to define areas on surfaces. This topic would be of greater interest if it were explored in more depth and defined in terms of its limitations and advantages. Mappings expressed without a reference to parameters or selection functions do not give value to the contribution

Thank you very much for the suggestion considered in the text.

No emphasis is placed on the need to have images taken orthogonally, perhaps with the use of a drone

 Thank you very much for the suggestion, very pertinent indeed, but at the time of the surveys it was not possible to have access to a drone, which would have been very useful and we would have been able to get detailed images.

The sentences in lines 36 and 93 are identical and therefore useless and repetitive at least one should be removed

 Thank you. We have deleted the redundant information.

The presence of pollution does not necessarily lead to conservation problems, as this depends on the environment, as also mentioned. There is a lack of a proposal for the possibility of benchmarks to understand how climate variation may affect or modify the conservation problems encountered

Thank you very much for the suggestion. Significantly improved the text of point 6 - Discussion of results and conclusions - that we reproduce here.

Climate variation introduces additional challenges and considerations for conservation efforts. Flexibility, adaptability, and a proactive approach to address emerging issues are crucial for successful conservation. Thus, we can propose some suggestions to understand how climate variation can affect or modify the conservation of this religious architectural heritage.

Changes in precipitation patterns, including more intense and frequent rainfall, can increase moisture levels. This excess moisture can lead to problems such as water infiltration, dampness, and mould growth.

Climate changes can influence the distribution and behaviour of pests and organisms representing a risk to cultural heritage. For example, warmer temperatures may increase insect activity, potentially accelerating the degradation of organic materials. On the contrary, drought periods can affect heritage stability [39].

The impacts of climate change can influence local communities and their relationships with cultural heritage, such as changes in traditional practices and economic activities that may affect the stewardship and care of the heritage.

Climate-related events can strain resources available for cultural heritage conservation. Emergency response efforts, recovery measures, and long-term maintenance may require additional funding and attention, additional resources, and the development of planned conservation projects.

Conservation practices may need to be adapted to address new challenges arising from climate change. Increased costs and efforts are required for adaptive conservation measures in order to achieve the sustainable development goals of the United Nations, namely the 13th goal: Climate Action by the year 2030 [40].

Addressing these challenges requires a multidisciplinary approach, incorporating climate science, architecture, and conservation practices to ensure the long-term sustainability of built architectural heritage to face climate change.

Cultural heritage conservation practices should incorporate climate resilience strategies to address these challenges. This may include developing climate-specific conservation plans, using sustainable materials and methods, using climate-resistant materials, and improving community engagement and awareness to ensure the continued protection of cultural heritage concerning changing climatic conditions. Collaboration among heritage conservation professionals, scientists, policymakers, and local communities is crucial to developing effective and sustainable solutions [39].

Round 2

Reviewer 2 Report

Comments and Suggestions for Authors

the modifications made do not explain how the alteration zones were identified and whether segmentation techniques were used

it is not clear whether areas described in a weighted manner are taken into account so that any increase in their prevalence can be verified

the tables are full of unnecessary terms and must to be simplified

interiors are not part of the assessment of external degradation and therefore not necessary in the context of the research

Author Response

Author's Reply to the Review Report ROUND 2- Reviewer 2

We appreciate the reviewer's suggestions and opinions, which, of course, contribute to improving our article. We have considered them, and we thank the reviewer for his or her time and effort.

It can be noted that all comments and suggestions have been considered, contributing to an improvement of the article.

Thank you very much.

the modifications made do not explain how the alteration zones were identified and whether segmentation techniques were used

Thank you very much for the suggestion—it is very pertinent indeed. However, at the time of the surveys, it was impossible to access a drone, which would have been useful for getting more detailed and higher-quality images.

However, very recently, we became aware that drones are prohibited in this area of the historic centre of Porto, which is classified as a World Heritage Site. So, our intention is to use the photogrammetry technique at the facades in the future, which will allow us to carry out a survey with much more detail and update our knowledge regarding the degradation problems and their evolution over time. This technique could be used several times in the future to monitor the evolution of the anomalies. Although we will also consider the methodology used in the present research, photographs will be used to compare them in other surveys with the ones in the present research and the future, and they will be taken from several angles, considering the facade and its details.

 it is not clear whether areas described in a weighted manner are taken into account so that any increase in their prevalence can be verified.

Thank you very much for the suggestion. In the future, we will consider using photogrammetry as regularly as possible over time to understand the evolution of the anomalies. The use of laser scanning monitoring techniques would also be useful, although it is dependent on the funding we can achieve to do that.  

 the tables are full of unnecessary terms and must to be simplified

Thank you very much for the suggestion. All tables have changed, but only the types of amendments have remained. Thus, all types of alterations not present in the facades of the churches studied were removed from the tables.

interiors are not part of the assessment of external degradation and therefore not necessary in the context of the research

Thank you very much for the suggestion that has been considered, and the description of the interior of the churches has already been substantially reduced. Cutting out all information about the interior of the churches will devalue the article. On the one hand, it is an added value in the sense of disseminating this Heritage, and it makes the article more attractive not only to specialists but also to other people interested in knowing this type of reality.

Round 3

Reviewer 2 Report

Comments and Suggestions for Authors

We thank the authors for their attention to the comments raised

It is suggested that more attention be paid to similar publications at an international level in order to understand the level of research and the possibilities for repeatable diagnosis and evaluation of alterations